# Blocking Muscarinic Receptor 3 Attenuates Tumor Growth and Decreases Immunosuppressive and Cholinergic Markers in an Orthotopic Mouse Model of Colorectal Cancer

**DOI:** 10.3390/ijms24010596

**Published:** 2022-12-29

**Authors:** Nyanbol Kuol, Majid Davidson, Jimsheena Karakkat, Rhiannon T. Filippone, Margaret Veale, Rodney Luwor, Sarah Fraser, Vasso Apostolopoulos, Kulmira Nurgali

**Affiliations:** 1Institute for Health and Sport, Victoria University, Melbourne 3011, Australia; 2Department of Physiology and Cell Biology, University of Nevada, Reno, NV 89557, USA; 3La Trobe Institute of Molecule Science, La Trobe University, Melbourne 3086, Australia; 4Royal Melbourne Hospital, University of Melbourne, Melbourne 3010, Australia; 5Immunology Program, Australian Institute of Musculoskeletal Sciences, Melbourne 3021, Australia; 6Department of Medicine Western Health, University of Melbourne, Melbourne 3010, Australia; 7Regenerative Medicine and Stem Cells Program, Australian Institute of Musculoskeletal Sciences, Melbourne 3021, Australia

**Keywords:** colorectal cancer, immunosuppressive, programmed death ligands 1 and 2, muscarinic receptor 3, α7 nicotinic acetylcholine receptor, choline acetyltransferase

## Abstract

Tumor cells have evolved to express immunosuppressive molecules allowing their evasion from the host’s immune system. These molecules include programmed death ligands 1 and 2 (PD-L1 and PD-L2). Cancer cells can also produce acetylcholine (ACh), which plays a role in tumor development. Moreover, tumor innervation can stimulate vascularization leading to tumor growth and metastasis. The effects of atropine and muscarinic receptor 3 (M3R) blocker, 1,1-dimethyl-4-diphenylacetoxypiperidinium iodide (4-DAMP), on cancer growth and spread were evaluated in vitro using murine colon cancer cell line, CT-26, and in vivo in an orthotopic mouse model of colorectal cancer. In the in vitro model, atropine and 4-DAMP significantly inhibited CT-26 cell proliferation in a dose dependent manner and induced apoptosis. Atropine attenuated immunosuppressive markers and M3R via inhibition of EGFR/AKT/ERK signaling pathways. However, 4-DAMP showed no effect on the expression of PD-L1, PD-L2, and choline acetyltransferase (ChAT) on CT-26 cells but attenuated M3R by suppressing the phosphorylation of AKT and ERK. Blocking of M3R in vivo decreased tumor growth and expression of immunosuppressive, cholinergic, and angiogenic markers through inhibition of AKT and ERK, leading to an improved immune response against cancer. The expression of immunosuppressive and cholinergic markers may hold potential in determining prognosis and treatment regimens for colorectal cancer patients. This study’s results demonstrate that blocking M3R has pronounced antitumor effects via several mechanisms, including inhibition of immunosuppressive molecules, enhancement of antitumor immune response, and suppression of tumor angiogenesis via suppression of the AKT/ERK signaling pathway. These findings suggest a crosstalk between the cholinergic and immune systems during cancer development. In addition, the cholinergic system influences cancer evasion from the host’s immunity.

## 1. Introduction

Colorectal cancer (CRC) is ranked third among the most commonly diagnosed cancers and is the second cause of cancer-related deaths worldwide [1,2,3]. Due to the complex nature of CRC and the lack of early clinical symptoms, it is often detected in the advanced stages. Despite the continued advancement in treatment technology, the 5-year survival rate of patients with metastatic disease remains less than 10% [4,5], partly due to the lack of specific markers for early diagnosis, leading to cancer progression and metastatic spread. Thus, it is indispensable to develop effective diagnostic and therapeutic approaches.

Resistance against cancer cells and their annihilation relies on the induction of cytotoxic CD8+ T cells and their differentiation into cytolytic and T helper-1 (Th1) cells. Cancer cells can avoid the host’s immune scrutiny by using several defensive mechanisms, including upregulation of immunosuppressive factors, such as programmed death-ligand 1 and/or 2 (PD-L1 and PD-L2), downregulation of major histocompatibility complex (MHC)-I and co-stimulatory molecules, secretion of angiogenic factors, such as vascular endothelial growth factor (VEGF) and platelet-derived growth factor receptor α (PDGFRα), the AXL receptor tyrosine kinase (AXL), anti-inflammatory cytokines, i.e., interleukin (IL)-10 and transforming growth factor-β (TGF-β), thus preventing activation of T cells, resulting in cancer invasion [6,7,8,9,10]. Cancer cells overexpress PD-L1 and/or PD-L2 on their surface, which upon binding to programmed death protein 1 (PD-1) expressed by activated CD8+ T cells, leads to their inhibition and/or apoptosis [11]. Interestingly, PD-L1 serves as an anti-apoptotic factor in cancer cells, leading to lysis resistance by CD8+ T cells and apoptosis [12].

The role of PD-L1 overexpression remains contradictory, with some papers reporting that its overexpression is associated with poor prognostic outcomes while others report better survival outcomes. For instance, high PD-L1 expression is associated with tumor metastasis, poor prognosis, and shorter survival in CRC patients [13,14]. Similarly, PD-L1 expression in stromal or tumor cells is inversely correlated with FOXP3+ cell density in CRC patients, further reinforcing the fundamental role in modulating regulatory T cells (Treg) in the tumor microenvironment [15]. More recently, in cohort studies of 181 CRC patients, PD-L1 expression was associated with high CD8+ tumor-infiltrating lymphocytes (TILs), *BRAF* mutation, microsatellite instability (MSI), lower frequency of *K-ras,* and poor prognosis [16]. In contrast, other studies have suggested that the expression of PD-L1 is associated with good patient survival outcomes. For example, PD-L1 expression correlates with high TIL infiltration and longer recurrence-free survival in breast cancer and pulmonary adenocarcinoma patients [17,18,19].

The role of PD-L2 in human cancers is not as well studied as PD-L1. Depending on the microenvironmental stimuli, PD-L2 is expressed by a number of immune and non-immune cells, such as T cells, dendritic cells, and macrophages [20]. In breast cancer patients, expression of PD-L2 correlates with overexpression of human epidermal growth factor receptor 2 (HER-2) and estrogen receptor (ER)-negative tumors, recurrence at distant sites, and younger patients’ age [19]. In CRC, PD-L2 expression is independently associated with worse overall survival [21].

In addition to forming an immunosuppressive microenvironment, compelling data suggests bi-directional signaling between the nervous system and the tumor microenvironment via the release of neurotransmitters, neuropeptides, and other factors implicating their influence on tumor development [22]. Neurotransmitters play an essential role in the activation of signaling pathways such as phosphoinositide 3-kinase (PI3K), mitogen-activated protein kinase (MAPK), and protein kinase B (AKT), which are related to cell proliferation and survival [22]. For example, the neurotransmitter acetylcholine (ACh) can stimulate CRC cell proliferation, invasion, vascularization, and migration by binding to muscarinic receptor 3 (M3R) through activation of the epidermal growth factor receptor (EGFR), PI3K, extracellular signal-regulated kinase (ERK)_1/2_ and AKT pathways as well as alpha 7 nicotinic receptor (α7nAChR) through activation of Janus kinase 2 (JAK2)/Signal transducer and activator of transcription 3 (STAT3) pathway [23,24,25,26,27,28,29]. In addition, cancer cells can also overexpress choline acetyltransferase (ChAT), a precursor enzyme required for ACh synthesis, and vesicular acetylcholine transporter (VAChT), essential for the transfer of ACh from the cytoplasm into synaptic vesicles [24,30].

Furthermore, the cholinergic nervous system plays an important role in tumor angiogenesis and metastasis [7,8]. For example, nicotine stimulation enhances VEGF expression and micro-vessel density in human colon cancer xenografts in nu/nu (nude) mice [31]. Administration of autoantibodies against muscarinic ACh receptors (mAChRs) in mouse models of breast cancer-mediated tumor angiogenesis via activation of mAChRs through VEGF-A release [32]. In addition, the administration of muscarinic agonist carbachol in the presence or absence of various muscarinic antagonists shows an increase in VEGF expression, as noted in LMM3 murine mammary adenocarcinoma-bearing BALB/c mice [33].

Overall, the synergism of these neuro-immune markers has yet to be explored as potential targets in CRC development. In this study, using murine colon cancer cell line and the orthotopic mouse model of CRC, we determined (i) the effect of blocking M3R on cancer cells and tumor growth; (ii) the expression of immunosuppressive, cholinergic, and angiogenic markers; and (iii) the presence of tumor-infiltrating immune cells.

## 2. Results

### 2.1. Effect of Blocking Muscarinic Receptors on Proliferation, Apoptosis, and Choline Production in CT-26 Cells

#### 2.1.1. Atropine Decreases CT-26 Cells Proliferation in a Dose Dependent Manner

The effect of blocking all muscarinic receptors on the proliferation of CT-26 cells was assessed using WST-1 assay following treatment with various concentrations of atropine at different time points. Atropine significantly decreased CT-26 cell proliferation at 1–8 h compared to 24–48 h (Figure 1A), with 300–1000 µM being the most significant. It appears that atropine decreases proliferation in a dose-dependent rather than a time-dependent manner. A trend of lower proliferation in a dose-dependent manner was noted at 24–48 h. To compare the effects of atropine and a selective M3R blocker, CT-26 cells were incubated with 4-DAMP and atropine for 8 h. No significant differences were noted between atropine and 4-DAMP, except in 600 and 700 µM doses (Figure 1B). However, there was a trend of 4-DAMP being less effective compared to atropine. In addition, cells were incubated with carbachol, which activates cholinergic receptors, and donepezil, which prevents the breakdown of ACh, for 8 h with various concentrations, and a dose curve was generated (Figure 1C). Incubation with a cholinergic agonist, carbachol, and acetylcholinesterase inhibitor, donepezil, reversed the effect of 4-DAMP and atropine. There was no significant difference between carbachol and donepezil at 50–200 µM. However, differences were noted at high doses (300–1000 µM). These findings further enforced the vital role of ACh in cellular proliferation.

In all subsequent experiments, cells were treated with 100 µM of atropine and 4-DAMP for 8 h as this concentration and time point induced prominent inhibition of cell proliferation and viability. Donepezil at 500 µM dose was applied for 8 h in all subsequent experiments.

#### 2.1.2. CT-26 Cells Can Produce the Choline Required for ACh Synthesis

Colon cancer cells can synthesize and release ACh [24,27,34,35]. To determine whether mouse colon cancer CT-26 cells could synthesize ACh, the amount of choline, a precursor for ACh, was measured in cell lysates (1 × 10^6^ cells). A choline/acetylcholine assay kit was used to measure the amount of choline in cell lysates. It was noted that CT-26 cells could produce a higher amount of choline when treated with donepezil compared to control, 4-DAMP, and atropine treatments (Figure 1D). No significant differences were observed between choline released in the cells treated with atropine and 4-DAMP compared to the control. Overall, atropine and 4-DAMP did not affect the CT-26 cells’ ability to produce choline; however, donepezil increased choline production.

#### 2.1.3. Atropine and 4-DAMP Induce Apoptosis in CT-26 Cells

Apoptosis is a natural cellular process that safeguards all body systems. Tumor cells must be resistant to anoikis (evasion from apoptosis), one of the essential steps in cancer metastasis [7]. Studies have demonstrated that several neurotransmitters, including ACh, play a significant role in regulating cell apoptosis [8]. To determine whether atropine and 4-DAMP induce apoptosis or necrosis in CT-26 cells, Annexin V and PI were used and analyzed by flow cytometry. Blocking all muscarinic receptors with atropine and M3R with 4-DAMP induced apoptosis in CT-26 cells compared to control (Figure 1E–E”). In fact, 20.1% of atropine-treated and 48.29% of 4-DAMP-treated cells underwent apoptosis compared to 10.76% of control cells.

### 2.2. Effect of Atropine and 4-DAMP on the Expression of Immunosuppressive and Cholinergic Markers in an In Vitro Model

#### 2.2.1. Atropine Decreases PD-L1 and PD-L2 Expression in CT-26 Cells

To determine whether atropine and 4-DAMP can influence the expression of PD-L1 and PD-L2, CT-26 cells were pre-treated with 100 µM atropine and 4-DAMP for 8 h prior to protein expression via Western blot staining. Atropine significantly decreased the expression of PD-L1 compared to the control (Figure 2A,B). Similarly, atropine attenuated PD-L2 expression compared to the control (Figure 2A,C). However, specific blocking of M3R with 4-DAMP did not affect CT-26 expression of PD-L1 and PD-L2 (Figure 2A–C).

#### 2.2.2. Atropine and 4-DAMP Attenuate M3R and ChAT Expression in CT-26 Cells

CT-26 cells produce choline, but the expression of ChAT, an enzyme required for ACh synthesis, and M3R were not determined. Therefore, the expression of ChAT and M3R in CT-26 cells and the effects of muscarinic receptor blockade on their expression were assessed. Western blot was utilized to evaluate the effects of atropine and 4-DAMP on CT-26 ability to express M3R and ChAT. The results show that CT-26 cells have a prominent expression of M3R and ChAT. Atropine and 4-DAMP treatments significantly reduced M3R expression (Figure 2A,D); however, did not affect ChAT expression (Figure 2A,E).

#### 2.2.3. Atropine and 4-DAMP Inhibit Phosphorylation of Kinases and Activation of EGFR in CT-26 Cells

There is a close link between M3R expression and phosphorylation of AKT and ERK and activation of EGFR [27]. Thus, the present study evaluated the effect of atropine and 4-DAMP on the phosphorylation of AKT and ERK and the activation of EGFR in CT-26 cells. The results showed that atropine inhibits the activation of EGFR and suppresses the phosphorylation of AKT and ERK in CT-26 cells (Figure 3A–D). Similarly, 4-DAMP suppresses phosphorylation of AKT and ERK (Figure 3A,C,D), whereas no significant effect on EGFR was observed (Figure 3A,B). These findings suggest that atropine exhibits its effect by inhibiting the EGFR/AKT/ERK pathway, while 4-DAMP exerts its effect via the suppression of the AKT/ERK signaling pathway.

### 2.3. Effect of 4-DAMP on the Tumor Growth, Expression of Immunosuppressive, Cholinergic, Angiogenic Markers and Tumor-Infiltrating Immune Cells in an In Vivo Model

#### 2.3.1. 4-DAMP Decreases Tumor Growth in an Orthotopic Mouse Model CRC

Mouse CT-26 colon cancer cells can form fast-growing and extremely vascularized tumors, making it useful for assessing the effects of therapeutic agents and their mechanisms of action [36]. To assess whether blocking M3R influences CRC tumor growth, CT-26 cells were implanted into the mouse caecum wall. Five days post-surgery, tumor-bearing mice were injected with either DMSO (vehicle solution) or 4-DAMP intraperitoneally daily for 28 days. Mice were weighed daily for 3 weeks and no significant difference was observed in the weight of mice treated with DMSO and 4-DAMP ((Figure 4A). Mice were culled, tumors removed, and weight, size, and volume were measured. 4-DAMP significantly attenuated tumor size compared to DMSO treatment (Figure 4B,C). There was a significant reduction in tumor weight in the 4-DAMP-treated compared to DMSO-treated mice (Figure 4D). In addition, treatment with 4-DAMP significantly decreased tumor volume compared to DMSO (Figure 4E). Furthermore, tumors around the caecum were counted and collected. Tumor-bearing mice treated with DMSO had more polyps and invasive tumors around the caecum compared to 4-DAMP-treated mice (Figure 4F–H, the difference between values (∆): −22.00 ± 4.40, *p* < 0.001).

#### 2.3.2. 4-DAMP Attenuates PD-L1 and Increases FOXP3 In Vivo

Studies identifying the role of PD-L1 expression in CRC have been controversial. Some studies associate PD-L1 expression with poor prognosis, whereas others with good prognosis [10,37,38]. Infiltration of immune cells within the tumor microenvironment has been highly implicated in the disease progression and prognosis. In addition, the expression of PD-L1 inversely correlates with FOXP3 in tumor samples from CRC patients [39]. Hence, PD-L1 and FOXP3, a marker used to label the regulatory T cells, were evaluated in tumor samples from mice with CT-26 cell-induced CRC. The results demonstrated that in vivo treatment of tumor-bearing mice with M3R blocker, 4-DAMP, significantly reduced the expression of PD-L1 compared to DMSO-treated controls (Figure 5A,D and Appendix A, ∆: −22.88 ± 0.80, *p <* 0.0001). These results concur with findings that FOXP3 inversely correlated with PD-L1 expression (Figure 5B,D and Appendix A, ∆: 15.00 ± 1.16, *p <* 0.0001).

In addition, the expression of PD-L2 was also evaluated. Although PD-L2 has a similar function to that of PD-L1, no significant difference was noted between the expression of PD-L2 in tumors from 4-DAMP-treated and DMSO-treated mice (Figure 5C,D, Appendix A, ∆: 0.38.88 ± 0.82, *p =* 0.6553). These findings were further confirmed by Western blot. The results obtained from in vitro model demonstrated that 4-DAMP attenuated PD-L2; however, this was not confirmed in vivo. These findings reinforced the influence of the tumor microenvironment on the expression of immunosuppressive markers.

#### 2.3.3. Correlation of PD-L1 Expression with Cholinergic Markers

It was hypothesized that the neurotransmitters, particularly ACh, might play a significant role in the induction of immunosuppressive markers such as PD-L1. In vitro studies in human colon cancer cells [40] and murine CT-26 cell lines confirmed this hypothesis. To further evaluate this hypothesis in vivo, tumor-bearing mice implanted with CT-26 cells were treated with either DMSO or 4-DAMP administered intraperitoneally daily for 28 days. PD-L1 was co-labeled with ChAT, a cholinergic enzyme crucial for ACh synthesis, and VAChT, a vesicular ACh transporter essential for packaging ACh into vesicles. Results demonstrated that 4-DAMP treatment significantly reduced the expression of PD-L1 (Figure 6A’,B’,C, ∆: −24.56 ± 3.15, *p* < 0.0001) as well as the expression of ChAT (Figure 6A”,B”,D, ∆: −10.23 ± 2.27, *p* < 0.001) compared to DMSO treatment. In addition, VAChT expression was attenuated in tumors from 4-DAMP-treated mice compared to DMSO-treated mice (Figure 6A’’’,B’’’,E, ∆: −5.82 ± 0.91, *p* < 0.05). Furthermore, PD-L1 was co-localized with ChAT and VAChT in tumors from DMSO-treated mice; however, after 4-DAMP treatment, this co-localization was abolished (Figure 6A’’’’,B’’’’). This reinforces the interaction between PD-L1 and cholinergic markers in CRC.

#### 2.3.4. 4-DAMP Treatment Augments α7nAChR and Attenuates M3R and ChAT Expression In Vivo

Although both ChAT and VAChT have been implicated in CRC, several studies have reported the important role of ACh receptors, especially M3R, in CRC progression. ACh binds to muscarinic M3R and nicotinic α7nAChR, stimulating CRC cell proliferation, angiogenesis, tumor growth, and metastasis [24,27,35,41]. In this study, ACh receptors were co-labeled with ChAT in tumors from DMSO and 4-DAMP-treated mice. 4-DAMP treatment induced a significant increase in α7nAChR expression (Figure 6F’,G’,H, ∆: 6.38 ± 0.34, *p* < 0.0001), however, the underlining mechanisms for this augmentation are unclear. On the other hand, 4-DAMP treatment induced a significant reduction in M3R expression (Figure 6F”,G”,I, ∆: −7.38 ± 0.31, *p* < 0.0001) compared to DMSO treatment. These findings were further confirmed by Western blot (Figure 6J).

#### 2.3.5. Effect of 4-DAMP Treatment on Tumor-Infiltrating Immune Cells

The prognostic value of PD-L1 or other immune checkpoint inhibitors is influenced by the profile of infiltrating immune cells within the tumor microenvironment. Therefore, to determine the profile of tumor-infiltrating immune cells, tumors from mice with CRC were collected into RPMI media. Tumors were then mechanically dissected into small pieces and incubated at 37 °C for 1 h with collagenase before commencing labeling with antibodies of interest. To characterize leukocyte populations in tumor samples from DMSO and 4-DAMP-treated mice, flow cytometry was used. Only viable cells were analyzed from scatter plots, and compensation was performed (when applicable) to prevent false-positive/false-negative results (Appendix A). The gating strategy for CD45+ cells was defined from single-cell doublets (Appendix A). Lymphocyte populations were gated from CD45+ cells (Appendix A). No significant differences were observed in CD45+ cells in tumors from DMSO and 4-DAMP-treated mice (Figure 7A, ∆: 2.17 ± 6.47, *p* = 0.7428). However, the results showed that 4-DAMP treatment significantly increased CD4+ T cell infiltration compared to DMSO treatment (Figure 7B, ∆: 4.58 ± 0.88, *p* < 0.001). Similarly, CD8+ T cell infiltration was significantly increased in tumors from 4-DAMP-treated compared to DMSO-treated mice (Figure 7C, ∆: 8.55 ± 2.13, *p* < 0.01). Moreover, no significant differences were noted in CD4+/CD8+ T cells ratio in tumors from 4-DAMP-treated compared to DMSO-treated mice (Figure 7D, ∆: 0.79 ± 2.16, *p* = 0.2315). Conversely, 4-DAMP treatment significantly attenuated infiltration of γδ T cells compared to DMSO (Figure 7E, ∆: −17.02 ± 1.067, *p* < 0.0001).

Furthermore, infiltration of B cells and eosinophils was gated from CD45+ cells. Results showed that the proportions of infiltrating B cells and eosinophils were significantly reduced in 4-DAMP-treated compared to DMSO-treated mice (Figure 7F,G). The proportions of all macrophages (defined as M0) and M2 phenotype macrophages were statistically increased in 4-DAMP-treated compared to DMSO-treated mice. However, M2 macrophages presented only a small fraction of all infiltrating M0 macrophages (Figure 7H). This suggests that most M0 macrophages could be exhibiting M1 phenotypes, which have an antitumor effect. In regard to neutrophil infiltration, as there are no specific markers to distinguish between N1 and N2, overall neutrophil infiltration was evaluated. Neutrophil infiltration was enhanced in tumors from 4-DAMP-treated compared to DMSO-treated mice (Figure 7I). We speculate that these infiltrated neutrophils are N1 phenotype as they exert an antitumor effect.

#### 2.3.6. Blocking M3R Decreases Tumor Angiogenic Markers

Tumor angiogenesis is a critical step in cancer growth and metastasis [7,8,42,43]. To evaluate the effect of blocking M3R on tumor angiogenesis, the expression of angiogenic markers, VEGF and CD31, was evaluated. The results demonstrated that the administration of 4-DAMP suppressed VEGF expression (Figure 8A’,B’,C,E; ∆: −16.75 ± 1.60, *p* < 0.0001). CD31 is a transmembrane glycoprotein expressed by endothelial cells and is used as a specific endothelial marker in paraffin sections. Thus, tumor neovascularization was evaluated in mice bearing CT-26 tumors by immunohistochemical labeling tumors with anti-CD31 antibodies in freshly fixed tumors. Treatment with 4-DAMP attenuated the expression of CD31 (Figure 8A”,B”,D; ∆: −13.85 ± 1.10, *p* < 0.0001). There was more intense CD31 staining observed in freshly fixed tumors from DMSO (Figure 8F,F’) compared with 4-DAMP-treated (Figure 8G,G’) mice, supporting the immunofluorescence results in freshly fixed samples.

In addition, tyrosine kinases in tumor tissue lysates were evaluated using a mouse Phospho-RTK Array. Tumor samples from each group were pooled, and the expression level of kinases was measured. The results showed that the DMSO group overexpressed platelet-derived growth factor receptor α (PDGFRα) and Axl tyrosine kinases (Figure 9A), both involved in cancer cell proliferation, invasion angiogenesis, and migration. Treatment with 4-DAMP significantly reduced expression levels of PDGFRα and abolished Axl (Figure 9B). Although TGF-β is not a marker for tumor angiogenesis per se, studies have demonstrated a strong association between tumor expression of TGF-β and tumor angiogenesis [44,45]. Administration of 4-DAMP significantly reduced TGF-β expression compared to DMSO treatment (Figure 9C–E, ∆: −13.85 ± 1.03, *p* < 0.0001). Taken together, these findings reinforced the important role of ACh in tumor angiogenesis, and M3 receptors could be a potential therapeutic target for the inhibition of angiogenesis.

#### 2.3.7. 4-DAMP Treatment Inhibits Phosphorylation of AKT and ERK In Vivo

ACh acting on M3R has been shown to trigger EGFR signaling to persuade the phosphorylation of AKT and ERK_1/2_ [27]. To determine the effect of blocking M3R with 4-DAMP on the phosphorylation of EGFR, STAT3, AKT, and ERK_1/2_ in vivo, Western blot analysis was used. The results demonstrated that 4-DAMP treatment inhibited ERK_1/2_ phosphorylation compared to DMSO (Figure 10A,C; ∆: −0.25 ± 0.07, *p* < 0.05). Similarly, 4-DAMP treatment induced a significant reduction in phosphorylation of AKT when compared to DMSO (Figure 10B,C; ∆: −0.26 ± 0.09, *p* < 0.05). However, both EGFR and pSTAT3 were not detected in tumor tissues; this could be due to the fast degradability of these proteins.

## 3. Discussion

The immune system plays a key role in the eradication of cancer cells. However, multiple mechanisms are responsible for suppressing the immune system in cancer, one of which is the expression of immune checkpoint inhibitors, including PD-1, PD-L1, and PD-L2 [6,10].

These molecules function by inhibiting the antitumor effects of T cell-mediated immune response. Although current therapies target some of these molecules, they have shortcomings, such as causing adverse events. Therefore, it is crucial to understand the possible mechanisms influencing the expression of these molecules. It was hypothesized that the cholinergic system might play a significant role in the induction of immunosuppressive markers such as PD-L1 and PD-L2. Indeed, the in vitro results demonstrated that CT-26 cells expressed PD-L1 and PD-L2, which were attenuated by cholinergic blockers, atropine, and 4-DAMP. To further evaluate whether the effects of 4-DAMP in the in vitro model could be achieved in the in vivo model of CRC, mice bearing CT-26 cell-induced tumors were injected daily with a vehicle, DMSO, or 4-DAMP for three weeks. It was clear that treatment with 4-DAMP decreased tumor weight, volume, and size compared to the vehicle-treated group. Furthermore, 4-DAMP treatment significantly decreased PD-L1 but not PD-L2 expression in the CRC mouse model. Furthermore, 4-DAMP administration significantly decreased cholinergic and angiogenic markers compared to DMSO treatment. In addition, 4-DAMP treatment augmented the antitumor immune response through increased infiltration of CD4+ and CD8+ T cells.

ACh is one of the main neurotransmitters found abundantly in the body. For a long time, it was believed that only neurons could synthesize and secrete ACh. However, studies have proven that many other cells, including tumor cells, can also produce ACh. In CRC, the expression of M3R plays an important role in cellular processes such as proliferation, differentiation, angiogenesis, invasion, metastasis, and the establishment of cell–cell contact [7,8] via the activation of various signaling pathways such as AKT, ERK, and EGFR [46]. In the present study, we evaluated whether blocking all muscarinic receptors with atropine and a selective M3R blocker, 4-DAMP, had a detrimental effect on murine CT-26 colon cancer cells. Blocking of all muscarinic receptors with atropine and M3R with 4-DAMP significantly suppressed CT-26 cell proliferation in a dose-dependent manner and induced apoptosis through the inhibition of EGFR/AKT/ERK signaling pathways. These findings concur with current literature implicating ACh acting on muscarinic receptors promotes cancer cell proliferation, invasion, and metastasis. For example, in human colon cancer cell lines SNU-C4, HT-29, and H508, administration of atropine, muscarinic receptors inhibitor eradicated SNU-C4 cell migration and HT-29 invasion; however, H508 cell migration entails the activation of MMP7 via EGFR and ERK signaling pathways [34,35]. Although blocking of M3R did not affect the activation of EGFR, as shown in this in vitro study, other studies have demonstrated a link between EGFR and M3R [27,47].

In addition, activation of nicotinic acetylcholine receptors (nAChRs) with nicotine facilitates cellular invasion and metastasis of human colon cancer cells, LOVO and SW620, via the activation of the p38 mitogen-activated protein kinase (MAPK) signaling pathway [48]. Likewise, nicotine stimulates the activation of α9-nAChR, which facilitates cellular migration of MDA-MB-231 and MCF-7 breast cancer cells via the expression of epithelial-mesenchymal transition markers [49]. Furthermore, in hepatocellular carcinoma, ACh acting on androgen receptor endorses SNU-449 cell invasion and migration via activation of signal transducer and activator of transcription 3 (STAT3) and AKT signaling pathways [50].

The in vivo data showed that blocking M3R with 4-DAMP in an orthotopic mouse model of CRC significantly reduced tumor weight, volume, and size compared to DMSO-treated controls. These findings are supported by previous studies demonstrating that administration of M3R antagonists, darifenacin, and 4-DAMP, significantly decreased tumor growth in a mouse model of gastric cancer [27]. Furthermore, 4-DAMP treatment significantly decreased PD-L1 expression in the CRC mouse model. However, no significant difference was observed in the expression of PD-L2 between DMSO and 4-DAMP treated groups. In addition, the administration of 4-DAMP significantly attenuated the expression of M3R and ChAT; nevertheless, there was an increase in α7nAChR expression. It was speculated that the possible mechanism involved in the attenuation of tumor growth by the 4-DAMP treatment could be through the inhibition of immunosuppressive (PD-L1) and cholinergic (M3R and ChAT) markers. Findings by Kamiya et al. (2019) in chemically induced and xenograft models of breast cancer demonstrated that presynaptic cholinergic neurostimulation resulted in decreased immune checkpoint molecules (PD-1 and PD-L1) expression and attenuated tumor growth [51]. Our findings provide evidence that blocking cholinergic receptors on a postsynaptic membrane reduced immunosuppressive markers and decreased tumor growth. Thus, in both cases, inhibition of immunosuppressive molecules had pronounced antitumor effects.

The tumor microenvironment is complex as it involves many factors, including resident and infiltrating immune cells, various stromal cells, and blood and lymphatic vessels, all of which are concealed in an extracellular matrix [52,53]. In fact, the nervous system and tumor microenvironment communicate through a feedback loop mechanism that facilitates tumor growth and progression [7,54]. There is also a complex interaction between immunosuppressive markers and the tumor microenvironment [6]. In the present study, the profile of tumor-associated immune cells was evaluated using flow cytometry cell sorting. Blocking of M3R significantly improved immune response against cancer, as noted by the increased expression of CD4+ and CD8+ T cells leading to reduced tumor size, weight, and volume. Furthermore, other infiltrating immune cells such as γδ T cells, B cells, and eosinophils, which have a deleterious effect on immune response, were abundantly decreased.

Tumor angiogenesis, essential for oxygen and nutrient supply, is one of the tumor traits promoting its growth. Studies have reported that tumors not only can form their own blood vessels but also produce neurotransmitters and immunosuppressive molecules such as PD-L1 and PD-L2 to promote tumor angiogenesis [8]. The present study showed that 4-DAMP treatment significantly decreased tumor angiogenesis, as demonstrated by the decreased expression of VEGF, CD31, and TGF-β. VEGF promotes the proliferation and expansion of endothelial cells via interaction with its receptors [55]. Studies have demonstrated elevated levels of VEGF and its receptors in human colon carcinomas and tumor-infiltrating endothelial cells [55,56,57]. Aside from its role in facilitating tumor angiogenesis, VEGF also exerts deleterious effects on the immune system by dampening a number of immune cells [58]. Studies have suggested that cholinergic signaling and the expression of immunosuppressive markers such as PD-L1 and PD-L2 play a functional role in tumor angiogenesis. For instance, the expression of PD-L1 correlates with VEGF, as noted in primary human glioma samples [59]. Similarly, the expression of PD-L1 and PD-L2 was positively associated with VEGF expression in renal carcinoma [60]. In CRC, high expression of VEGF and CD31 was correlated with poor patient survival [61]. Expression of immune checkpoint molecule B7-H3 correlates with CD31 expression in tissue samples from patients with CRC and induced VEGFA through the activation of the NF-κB pathway as observed in vitro and in vivo [62].

TGF-β acts as an anti-tumorigenic factor in the early stages and a pro-tumorigenic factor in the late stages of tumor progression [63]. High levels of TGF-β in the tumor microenvironment have been associated with angiogenesis, contributing to tumor development and metastasis [44,45]. In human gastric cancer cell lines, MKN45 and KATOIII, TGF-β1 induced VEGF-C expression leading to lymphangiogenesis by activating Smad2/3 and Smad pathway [64]. In addition, 4-DAMP significantly attenuated PDGFRα and abolished Axl, both of which play an essential role in tumor angiogenesis. In tumor specimens from CRC patients, PDGFRα/β expression correlates with lymphatic dissemination and metastasis [65,66]. Similarly, in invasive ductal carcinoma, overexpression of PDGFRα correlates with metastasis [67]. Axl promotes the survival of endothelial cells and the remodeling of endothelial barriers in wound healing and vessel impairment [68]. Axl is essential for angiogenesis mediated by VEGF-A activation of VEGFR-2 via the PI3K/AKT pathway [69]. In in vitro and in vivo models of breast and prostate cancers, inhibition of Axl suppressed pro-angiogenic factors and impaired functional properties of endothelial cells [70].

The results of this study demonstrated that enhanced expression of PD-L1, PD-L2, M3R, and ChAT and angiogenic markers were attenuated by treatments with cholinergic receptor blockers in vitro. In vivo results demonstrated that blocking M3R has pronounced antitumor effects via several mechanisms, including inhibition of immunosuppressive molecules, enhancement of antitumor immune response, and suppression of tumor angiogenesis via suppression of the AKT/ERK signaling pathway.

## 4. Materials and Methods

### 4.1. Mice

Male BALB/c mice aged 5–8 weeks (n = 16) were purchased from the Animal Resources Centre and housed in groups of 4 per cage. All animals were kept in a temperature-controlled environment with a 12 h light/dark cycle at approximately 22 °C with access to food and water. The mice were allowed to acclimate for at least one week before surgery. All animal experiments in this study complied with the guidelines of the National Health and Medical Research Council (NHMRC) Australian Code of Practice for the Care and Use of Animals for Scientific Purposes under the approval of the Victoria University Animal Experimentation Ethics Committee (ethics number AEETH 15-011). All efforts were made to lessen animal suffering.

### 4.2. Cell Culture

Murine colorectal cancer cell line (CT-26) was obtained from the American Tissue Culture Collection (ATCC, Catalogue no. CRL-2638), Manassas, VA, USA. CT-36 cell was cultured in Roswell Park memorial institute (RPMI) 1640 culture media supplemented with 10% fetal bovine serum, 1% penicillin-streptomycin, and 1% Glutamine, at 37 °C, in 5% CO_2_ and 95% air atmosphere. Passage of cells was conducted with 0.25% trypsin and 0.02% ethylenediamine tetraacetic acid (EDTA) every 3–4 days. When cells grew into confluent or semi-confluent monolayers in the 75 cm^2^ medium flasks, they were either passaged or used.

### 4.3. Cell Viability

The water-soluble tetrazolium-1 (WST-1) assay kit (Roche Diagnostics GmbH, Penzberg, Germany) was used to determine the viability of CT-26 cells. WST-1 is cleaved to form formazan dye via a complex cellular interaction at the cell surface. This interaction is contingent on the viable cells’ production of glycolytic nicotinamide adenine dinucleotide phosphate (NADPH). Hence, the formed formazan dye correlates to the number of viable cells in the culture. CT-26 cells were seeded and cultured at 1 × 10^4^ cells per well in 96 well plates for 24 h. Cells were then treated with various concentrations of the general muscarinic receptor blocker, atropine (Sigma-Aldrich, Australia), selective M3R blocker, 1,1-dimethyl-4-diphenylacetoxypiperidinium iodide (4-DAMP) (Abcam, Adelaide, SA, Australia), cholinergic agonist, carbachol (Abcam, Australia) and acetylcholinesterase inhibitor, donepezil (Abcam, Australia) for 8 h. All treatments were performed in triplicates, and three independent experiments were conducted. WST-1 reagent (10 µL) was added to each well and incubated at 37 °C for 1 h. Cellular proliferation at the absorbance of 450 nm was measured using a microplate reader (Varioskan Flash, Thermo Fisher Scientific, Scoresby, VIC, Australia).

### 4.4. Annexin V Apoptosis Assay

CT-26 cells were cultured overnight in six-well plates at 5 × 10^5^ cells per well. Cells were treated with 100 µM of atropine and 4-DAMP for 8 h. Cells were collected and resuspended in fluorescence-activated cell sorting (FACS) buffer and labeled with 100 µL per well with Annexin V at 1:1000 dilution and 0.5 µg/mL of propidium iodide (PI). Flow cytometry was utilized to determine the percentage (%) of apoptotic and necrotic cells. All treatments were performed in duplicates, and two independent experiments were conducted.

### 4.5. Choline/Acetylcholine Assay

The choline/acetylcholine assay kit (Abcam, Australia) was used to measure choline concentration in CT-26 cell lysates. The assay was carried out in accordance with the instructions provided by the manufacturer. Briefly, CT-26 (1 × 10^6^) cells were cultured overnight, after which cells were treated with 100 µM of cholinergic antagonists, atropine, and 4-DAMP, and 500 µM of acetylcholinesterase inhibitor, donepezil. Cells were lysed in 500 µL choline assay buffer before commencing choline measurements using a microplate reader (Varioskan Flash, Thermo Scientific, Australia) at the absorbance of 570 nm. All treatments were performed in duplicates, and two independent experiments were conducted.

### 4.6. Orthotopic Implantation of CT-26 Tumor Cells

Mice were anesthetized using xylazine (10 mg/kg) and ketamine (80 mg/kg) injected intraperitoneally. The level of anesthesia during the surgery was monitored using the paw pinch reflex test. The eyes of the animals were treated with ViscoTears to protect them from drying out during the surgery. Mice were placed on an operating table on a heat mat (30–36 °C), and all procedures were performed under aseptic conditions. All instruments were autoclaved and only opened when ready to operate. The abdomen was shaved, swabbed with 70% ethanol, and covered with sterile film. A small midline abdominal incision was made, and the caecum was exteriorized on sterile gauze. Matrigel (25 μL) containing CT-26 cell suspension (1 × 10^6^ cells) was injected into the caecum wall of BALB/c mice using an insulin needle. After injection, the abdominal muscle wall was sutured using polygalactone, and the skin using surgical silk or dissolvable skin sutures. The incision area was sterilized with saline, followed by iodine. Mice were given an analgesic Temgesic/Buprenorphine (0.05 mg/kg) subcutaneously. Mice were then monitored visually during recovery time (about 1–1.5 h), and when fully conscious, they were returned to an animal holding room in the animal facility.

### 4.7. Intraperitoneal Injections and Tissue Collection

After five days post-surgery, vehicle BALB/c mice received an intraperitoneal injection of 0.1% dimethyl sulfoxide (DMSO) treatment and the study group received 10 mg/kg of 1,1-dimethyl-4-diphenylacetoxypiperidinium iodide (4-DAMP) per a day [27]. The volume of the administered solution was calculated per body weight with a maximum volume of 200 μL per injection. Mice were culled at 28 days post-surgery via lethal injection of phenobarbital, and tumors were removed, weighed, and used for Western blot, flow cytometry, proteome profiler array, and immunohistochemistry. Tumor tissues were used to assess angiogenesis, tumor-infiltrating immune cells, and immunosuppressive and cholinergic markers’ expression. Tumor tissues used for flow cytometry analysis were collected into RPMI media, western and proteome profiler arrays were snap-frozen in liquid nitrogen, and samples used for immunohistochemistry were placed in Zamboni’s fixative (2% formaldehyde 0.2% picric acid).

### 4.8. Immunohistochemistry in Cross-Sections

Tumor tissues collected from vehicle-treated and 4-DAMP-treated groups were fixed with Zamboni’s fixative overnight at 4 °C. The next day, the fixative was cleared off by washing samples for 10 min three times with DMSO (Sigma-Aldrich, Macquarie Park, NSW, Australia), followed by three times 10 min wash with phosphate-buffered saline (PBS). Tissues were then embedded in an optimum cutting temperature medium (OCT) and frozen using 2-methyl butane (isopentane) and liquid nitrogen. Samples were stored in a −80 °C freezer until cryo-sectioned. Tissues were cut at 10 μm section thickness using a Leica CM1950 cryostat (Leica Biosystems, Nussloch, Germany), adhered to slides, and allowed to dry at room temperature for 1 h before commencing the staining process. OCT was washed off with PBS containing 0.01% Triton X-100 (PBST) for 5 min. Samples were outlined using a liquid Blocker Super Pap Pen to reduce the volume of antibodies used. The endogenous activity was blocked using 10% normal donkey serum for 1 h at room temperature, followed by PBST washes. Samples were then incubated with 1:500 dilution of primary antibodies [mouse monoclonal to PD-L1 (Abcam, ab210931), rabbit polyclonal to PD-L2 (Abcam, ab200377), rabbit polyclonal to M3R (Abcam, ab126168), mouse monoclonal to α7nAChR (Novus, Centennial, CO, USA, 7F10G1), goat polyclonal to ChAT (Abcam, ab134021), sheep polyclonal to VAChT (Abcam, ab31544), mouse monoclonal to FOXP3 (Abcam, ab20034), rabbit polyclonal to VEGF (Abcam, ab46154), rat monoclonal to CD31 (Abcam, ab7388) and rabbit polyclonal to TGF-β (Abcam, ab155264)] overnight at 4 °C. Sections were then washed in PBST before incubation with 1:250 dilution of secondary antibodies labeled against primary antibodies [Alexa Fluor 488-conjugated donkey anti-goat, anti-sheep, and anti-rat (Abacus, Morningside, QLD, Australia, JI705-545-003), Alexa Fluor 594-conjugated donkey anti-rabbit (Abacus, JI711585152) and Alexa Fluor 647-conjugated donkey anti-mouse (Abacus, JI715605150)] for 2 h at room temperature in the dark, followed by PBST washes. The sections were incubated with 4′,6-diamidine-2′-phenylindole dihydrochloride (DAPI) (D1306, Life Technologies, Mulgrave, VIC, Australia) for 1 min. Sections were given final washes in PBST and then mounted with DAKO mounting media (Agilent Technologies, Mulgrave, VIC, Australia). Coverslips were placed over each section and left to dry overnight before imaging. Sections were viewed under a Nikon Eclipse Ti laser scanning confocal microscope (Nikon, Tokyo, Japan), whereby eight randomly chosen images from each sample were captured with a 40× objective and analyzed using image analysis software (Nikon, Japan).

### 4.9. Immunoperoxidase Staining

Tumor samples were collected and placed in Zamboni’s fixative overnight. The endogenous activity was blocked using 3% hydrogen peroxide for 30 min. Samples were incubated overnight with primary antibody against CD31 (1:100). After PBST washes, samples were incubated at room temperature for 2 h with secondary antibodies at 1:250 dilution. CD31 antibody was diluted in 0.5% bovine serum albumin (BSA)-RPMI. Samples were incubated with developer reagent 3,3′-diaminobenzidine tetrahydrochloride (DAB) liquid substrate in peroxidase buffer for 30 min. Samples were counter-stained with Mayer’s Haematoxylin, rinsed in tap water, Scott’s tap water, 100% ethanol, and xylene. Samples were mounted in an aqueous mounting medium and imaged using slide scanning imaging systems.

### 4.10. Western Blot

Proteins extracted from CT-26-induced tumor tissues and CT-26 cells were evaluated for the expression of immunosuppressive, cholinergic, and angiogenic markers as well as cell signaling pathways, epidermal growth factor receptor (EGFR), phospho extracellular signal-regulated kinase (pERK_½_) and phospho serine/threonine kinase or protein kinase B (pAKT) by Western blot. CT-26 cells were incubated with 100 µM atropine and 4-DAMP for 8 h. After treatments, cells were collected and lysed in radioimmunoprecipitation assay (RIPA) buffer (pH 7.4, 150 mM NaCl, 0.1% SDS, 0.5% sodium deoxycholate, 1% NP-40 in PBS, Sigma) containing protease and phosphatase inhibitors cocktail (Roche Applied Science, Switzerland). For tumor samples, 100 mg of tumor tissues per mouse were weighed, tumor samples from 3 mice per band for the first two bands, and tumor samples from 2 mice for the third band were pooled together. Samples were then homogenized in 500 µL of RIPA buffer containing protease and phosphatase inhibitors.

Cellular proteins (20 µg) from CT-26 cell line and 25 µg protein from tumor samples were separated by 8% to 12% sodium dodecyl sulfate (SDS)/polyacrylamide gel electrophoresis. The separated fragments were transferred to 0.22 µm polyvinylidene fluoride membranes, which were blocked with 5% skim milk in PBS containing 0.1% Tween 20 and incubated overnight at 4 °C in a platform shaker at 40 rpm speed. The membranes were incubated with immunosuppressive, cholinergic, and angiogenic markers, as well as cell signaling pathways, at dilution of 1:1000, rabbit monoclonal EGFR (Cell Signaling, Wangara, WA, Australia, #4267), rabbit monoclonal to pERK_½_ (Cell Signalling, Australia, #3192) and rabbit monoclonal to pAKT (Cell Signalling, Australia, #4060) overnight at 4 °C. The membranes were then incubated with HRP-conjugated secondary antibodies at a dilution of 1:10,000 [anti-mouse (Abcam, ab6789), anti-rabbit (Abcam, ab6721), and anti-goat (Abcam, ab6885)] for 2 h at room temperature followed by three times PBS-0.1% Tween 20 washes. Glyceraldehydes-3-phosphate dehydrogenase (GADPH) was used as a loading control. Protein detection was performed using enhanced chemiluminescence reagents. A chemiluminescent signal was captured using the FluorChem FC2 system. The expression level of each protein was quantified using ImageJ software.

### 4.11. Proteome Profiler Mouse Phospho-RTK Array Kit

The Proteome Profiler Mouse Phospho-RTK Array Kit is a membrane-based immunoassay that captures antibodies spotted in duplicate on nitrocellulose membranes binding to specific target proteins in the sample. The assay was carried out in accordance with the instructions provided by the manufacturer. Briefly, tumor samples from each group were pooled and lysed in Lysis Buffer 17 prepared with protease inhibitors. Samples were mixed by pipetting up and down to resuspend, and lysates were gently rocked at 4 °C for 30 min on a rocking platform shaker. Tumor lysates were centrifuged at 1500 rpm for 5 min at 4 °C, and supernatants were transferred into the clean test tubes. Array membranes were placed onto a 4-well multi-dish and incubated with Array Buffer 1 for 1 h at room temperature on a rocking platform shaker. After 1 h incubation, Array Buffer 1 was aspirated out, and membranes were incubated with tumor lysates overnight at 4 °C on a rocking platform shaker. Membranes were then washed with 1× Wash Buffer for 3 × 10 min. Membranes were incubated with anti-phosphotyrosine-HRP antibodies at room temperature for 2 h on a rocking platform shaker. Membranes were then washed with 1× Wash Buffer for 3 × 10 min. Membranes were then incubated with a chemiluminescence reagent mix, and a chemiluminescent signal was captured using the FluorChem FC2 system.

### 4.12. Flow Cytometric Cell Staining

On the day of culls, tumors were collected into RPMI media, and tumor tissues were processed into single-cell suspensions for FACS analysis. Single-cell suspensions were performed by mechanically dissecting tumors into small pieces and incubating with 2 mL of collagenase (0.1% *w*/*v* in 1 mL of α-MEM) at 37 °C for 2 h with 30 min intervals of mechanical dissociation. Tumor suspensions were filtered with 40 µm cell strainers Falcon^®^ into 50 mL Falcon^®^ tubes and were then centrifuged at 1500 rpm for 5 min at 4 °C. Cell pellets were incubated with 1× red blood cell lysing buffer for 3 min at 37 °C. Cell pellets were then resuspended in 1 mL of FACS buffer to create a single-cell suspension and accounted using a hemocytometer.

Viable cell pellets were incubated with two different antibody cocktails. Cocktail 1 contained leukocyte infiltration markers (FIT C anti-CD45, PE-Cy7 anti-CD11b, BV480 anti-CD4, APC-Cy7 anti-CD8a, AF647 anti-CD193(CCR3), BV421 anti-Siglec-F and Fc Block), while cocktail 2 was comprised of (FIT C anti-CD45, PE-Cy7 anti-CD11b, APC-Cy7 anti-CD19, AF647 anti-CD206, BV480 anti-CD115, PE anti-F480, PE-CF594 anti-Ly-6C, BV421 anti-Ly-6G and Fc Block). Tumor cells (10 × 10^6^) cells (400 µL) were aliquoted in BD Falcon^®^ FACS tubes. Cells were centrifuged at 1300 rpm for 3mins at 4 °C. Cells were then incubated with 200 µL of antibody cocktails for 1hr at 4 °C. After incubation, cells were centrifuged at 1300 rpm for 3 min at 4 °C and supernatants aspirated. Cells were resuspended in 200 µL FACS buffer and filtered through 35 µm filters in a 5 mL BD Falcon^®^ tube. Prior to FACS analysis, cells were incubated with viability solution, 7-amino-actinomycin D (7-AAD, 1:20), to gate the viable cell populations.

### 4.13. Data Analysis

Images were captured on a Nikon Eclipse Ti multichannel confocal laser scanning system (Nikon, Japan). Z-series images were acquired at a nominal thickness of 1 µm (1024 × 1024 pixels). Image J software (National Institute of Health, Bethesda, MD, USA) was employed to convert images from RGB to greyscale 8-bit binary; particles were then analyzed to obtain the percentage area of immunoreactivity [71]. All immunohistochemistry images and Western blot bands were quantified, and statistical analysis was performed by Student’s *t*-test.

The apoptosis analysis was performed and quantified by flow cytometry (BD FACS Canto II, BD Bioscience, San Jose, CA, USA) with the aid of BD FACs DIVA software version 6 (BD Bioscience, CA, USA). The characterization of tumor-infiltrating immune cells was performed and quantified by flow cytometry (BD FACS Aria III, BD Bioscience, CA, USA) with the aid of BD FACs DIVA software version 8 (BD Bioscience, CA, USA). Two hundred thousand events were collected per sample and analyzed by BD FACS DIVA software. Each antibody conjugate produces a distinctive emission spectrum, and each experiment employed single-color compensation controls to optimize photo-multiplier tube (PMT) voltages and calculate spectral overlap (where applicable). Excel, SPSS, and Prism (GraphPad Software, La Jolla, CA, USA) were used to aid the statistical analysis, and *p* < 0.05 was considered significant.

## Figures and Tables

**Figure 1 ijms-24-00596-f001:**
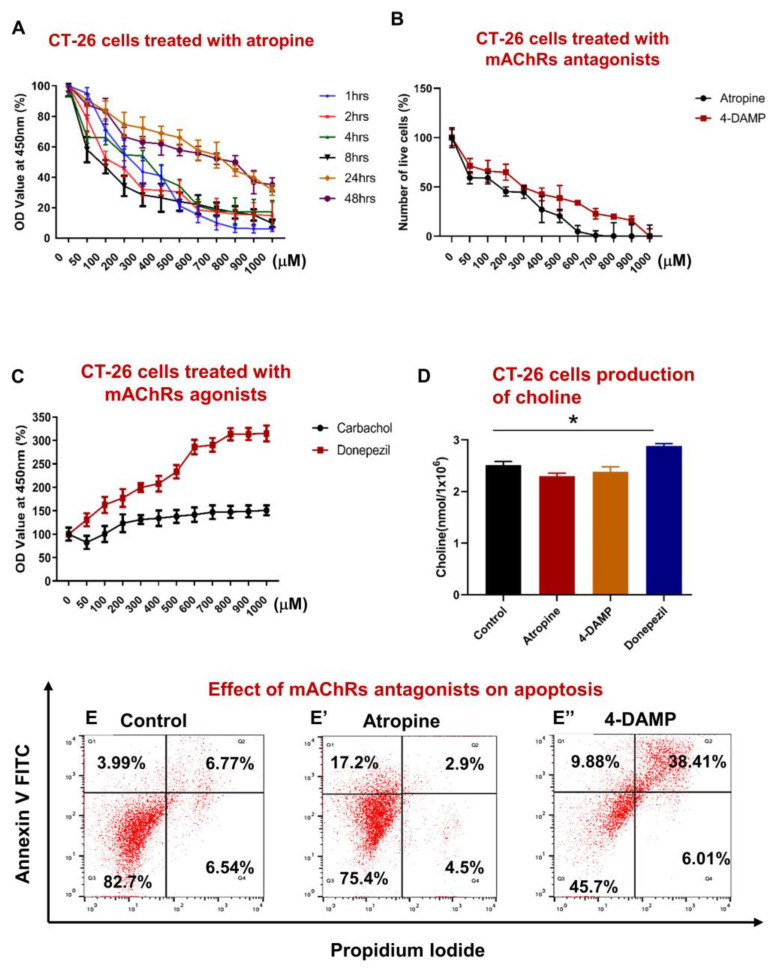
Effect of atropine and 4-DAMP on proliferation, apoptosis, and production of choline in CT-26 cells. CT-26 cells were treated with different concentrations of atropine at different time points (**A**). Number of viable cells after 8 h incubation with various concentrations of atropine and 4-DAMP (**B**). CT-26 cells were treated with various concentrations of carbachol and donepezil for 8 h (**C**). The amount of choline was measured in CT-26 cells treated with atropine, 4-DAMP, and donepezil (**D**). Annexin V-FITC/PI staining of CT-26 murine colon cancer cell line treated with control (**E**), atropine (**E’**), and 4-DAMP (**E”**). Values in (**A**–**E**) are mean ± standard error of the mean (SEM) from at least three independent experiments performed in triplicate wells. Two independent experiments were performed in triplicates. Values presented as mean ± SEM from at least two independent experiments for choline assay. Two-way ANOVA followed by Tukey’s multiple comparison test was used and considered significant when ** p* < 0.05.

**Figure 2 ijms-24-00596-f002:**
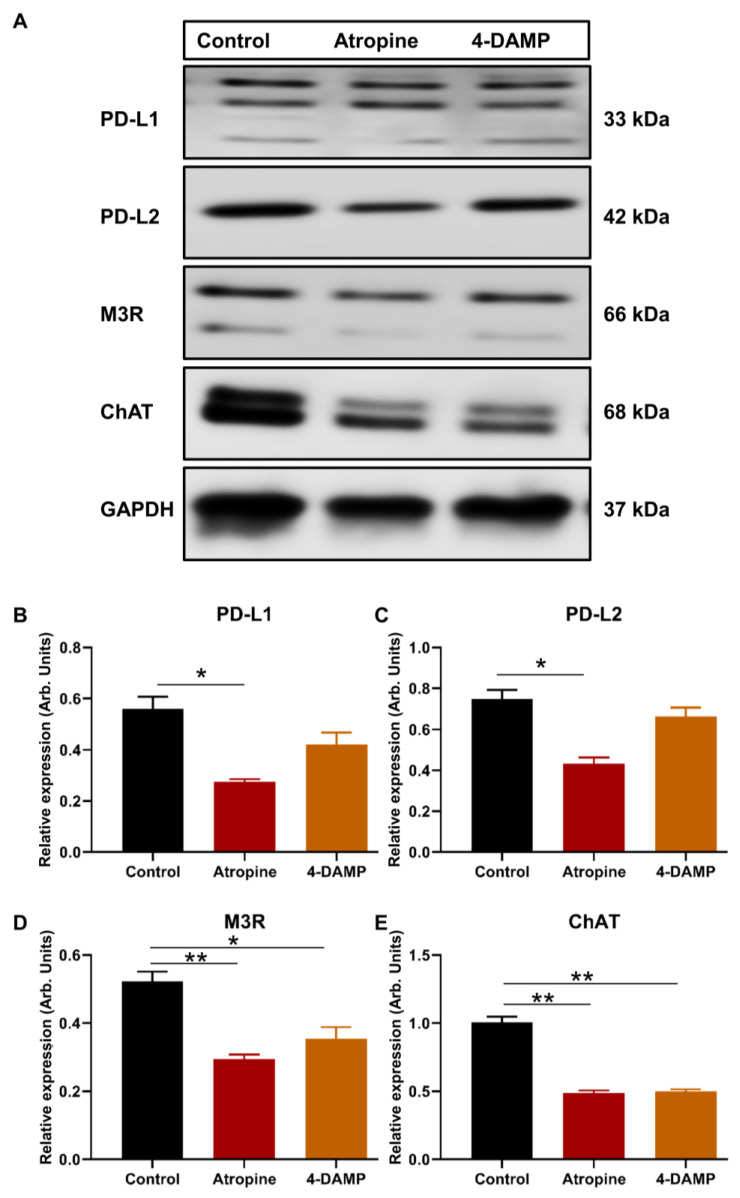
Effect of atropine and 4-DAMP on the expression of immunosuppressive and cholinergic markers in CT-26 cells. Western blot bands for PD-L1 and PD-L2 expression in CT-26 cells treated with atropine and 4-DAMP (**A**). Western blot bands for M3R and ChAT expression in CT-26 cells treated with atropine and 4-DAMP (**A**). Bar graphs displaying the mean intensity of PD-L1 (**B**) and PD-L2 (**C**) expression in CT-26 cells treated with 4-DAMP and atropine. Bar graphs displaying the mean intensity of M3R (**D**) and ChAT (**E**) expression in CT-26 cells treated with 4-DAMP and atropine. Data presented as mean ± SEM. Two-way ANOVA, * *p* < 0.05, ** *p* < 0.01. Data presented as mean ± SEM. Two-way ANOVA, * *p* < 0.05.

**Figure 3 ijms-24-00596-f003:**
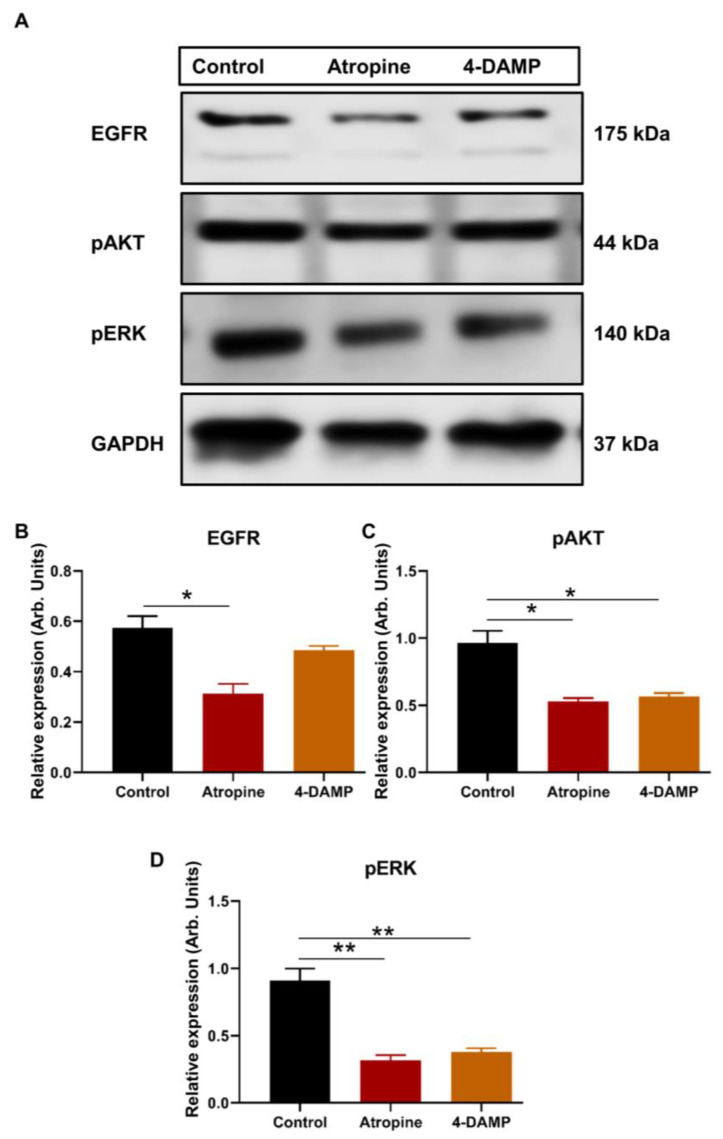
Effect of atropine and 4-DAMP on the expression of protein kinases in CT-26 cells. Western blot bands for EGFR, pAKT, and pSTAT3 expression in CT-26 cells treated with atropine and 4-DAMP (**A**). Bar graphs displaying the mean intensity of EGFR (**B**), pAKT (**C**), and pERK (**D**) expression in CT-26 cells treated with 4-DAMP and atropine. Data presented as mean ± SEM. Two-way ANOVA, * *p* < 0.05, ** *p* < 0.01.

**Figure 4 ijms-24-00596-f004:**
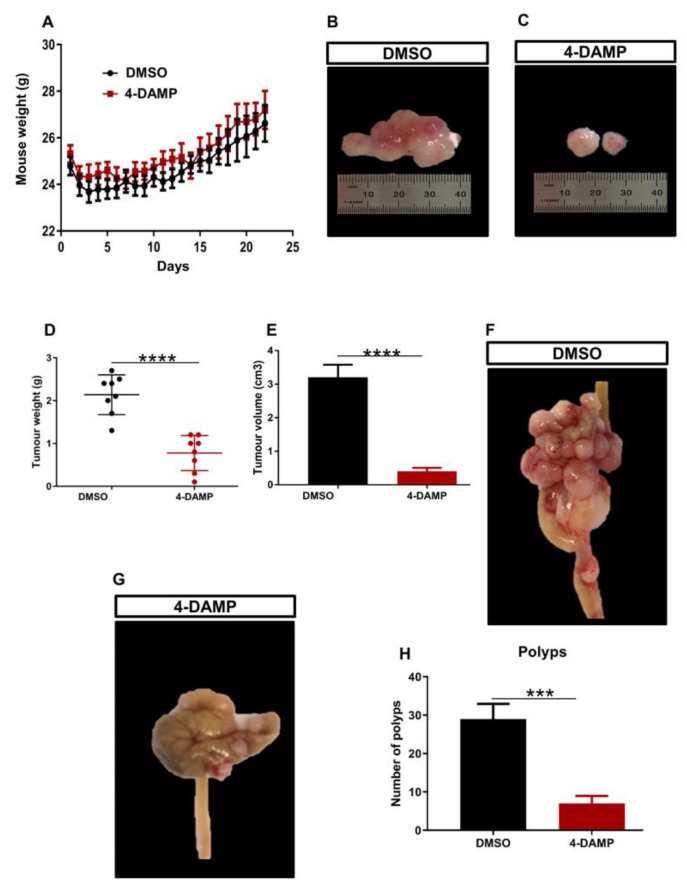
Effect of 4-DAMP on tumor growth in vivo. Body weight of mice treated with a vehicle (0.1% DMSO) and 4-DAMP (10 mg/kg daily) (**A**). Images of tumor size from DMSO-treated group (**B**) and 4-DAMP-treated group (**C**). Bar graphs displaying the mean weight (**D**) and volume (**E**) of tumors collected from DMSO and 4-DAMP-treated groups. The caecum samples were removed from tumor-bearing mice treated with DMSO (**F**) and 4-DAMP (**G**). Bar graph displaying the mean number of tumor polyps from DMSO and 4-DAMP-treated groups (**H**). Data presented as mean ± SEM, n = 8 mice per group. Student’s *t*-test, *** *p* < 0.001, **** *p* < 0.0001.

**Figure 5 ijms-24-00596-f005:**
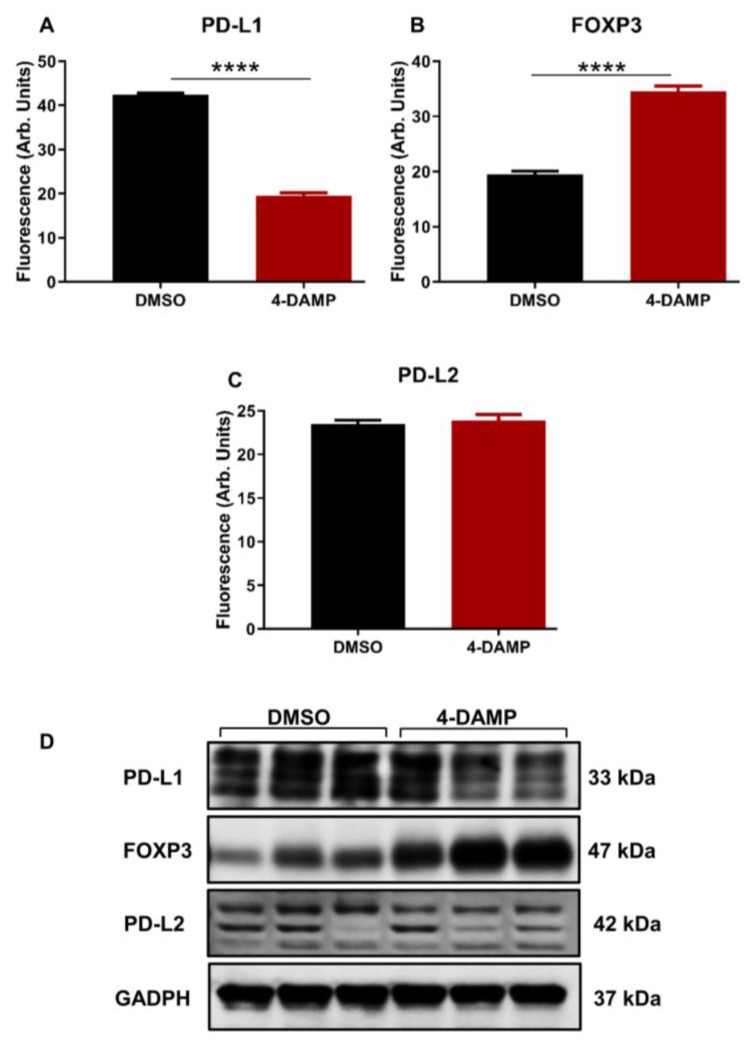
Effect of 4-DAMP on the expression of PD-L1 and FOXP3 in vivo. Bar graphs displaying the mean fluorescence of PD-L1 (**A**), FOXP3 (**B**), PD-L2 (**C**), and images of Western blot bands (**D**) in tumor samples from DMSO-treated and 4-DAMP-treated mice. Data presented as mean ± SEM, n = 8 mice per group. Student’s *t*-test, **** *p* < 0.0001.

**Figure 6 ijms-24-00596-f006:**
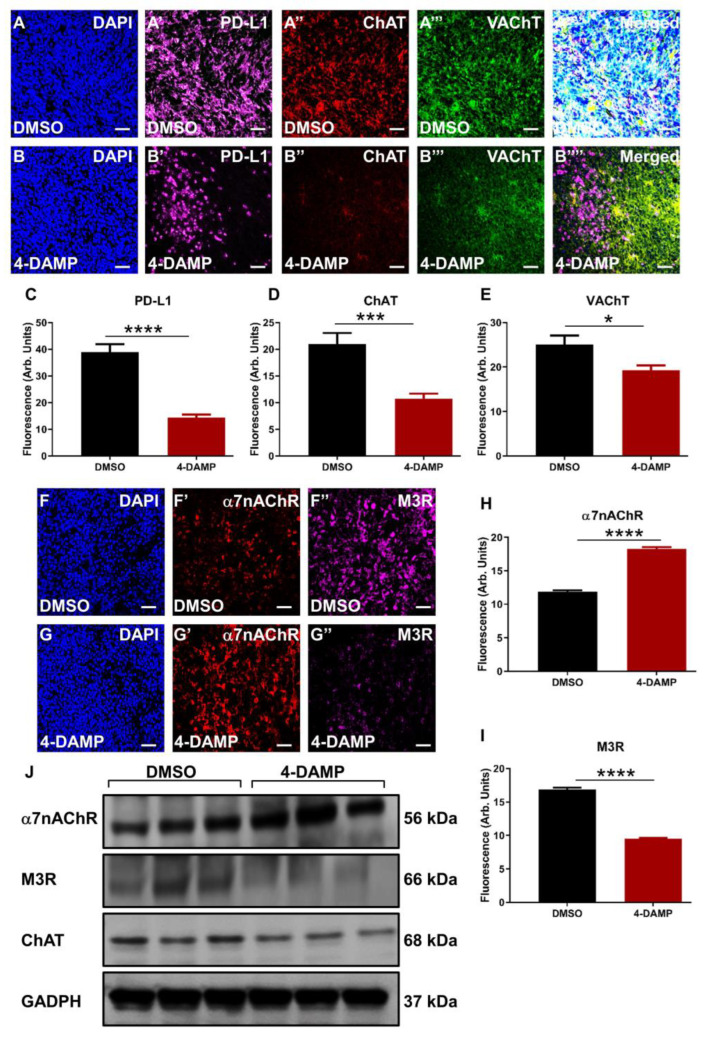
Correlation of PD-L1 expression with cholinergic markers in tumor samples from mice bearing CT-26 cell-induced tumors. Expression of PD-L1 and cholinergic markers (ChAT and VAChT) in tumor samples from mice bearing CT-26 cell-induced CRC treated with DMSO (**A**–**A’’’’**) and 4-DAMP (**B**–**B’’’’**). Tumors were labeled with the nuclei marker DAPI (blue; A-B), PD-L1 (magenta; A’-B’), ChAT (red; **A’’**,**B’’**), VAChT (green; **A’’’**,**B’’’**), and all markers merged (**A’’’’**,**B’’’’**). Bar graphs displaying the mean fluorescence of PD-L1 (**C**), ChAT (**D**), and VAChT (**E**) in tumor samples from DMSO-treated and 4-DAMP-treated mice. Expression of cholinergic markers in tumor samples from mice bearing CT-26 cell-induced CRC treated with DMSO (**F**–**F’’**) and 4-DAMP (**G**–**G’’**). Tumors were labeled with the nuclei marker DAPI (blue; **F**,**G**), α7nAChR (red; **F’**,**G’**), and M3R (magenta; **F’’**,**G’’**). Bar graphs displaying the mean fluorescence of α7nAChR (**H**), M3R (**I**), and Western blot bands (**J**) in tumor samples from DMSO-treated and 4-DAMP-treated mice. The scale bar represents 50 µm. Data presented as mean ± SEM, n = 8 mice per group. Student’s *t*-test, * *p* < 0.05, *** *p* < 0.001, **** *p* < 0.0001.

**Figure 7 ijms-24-00596-f007:**
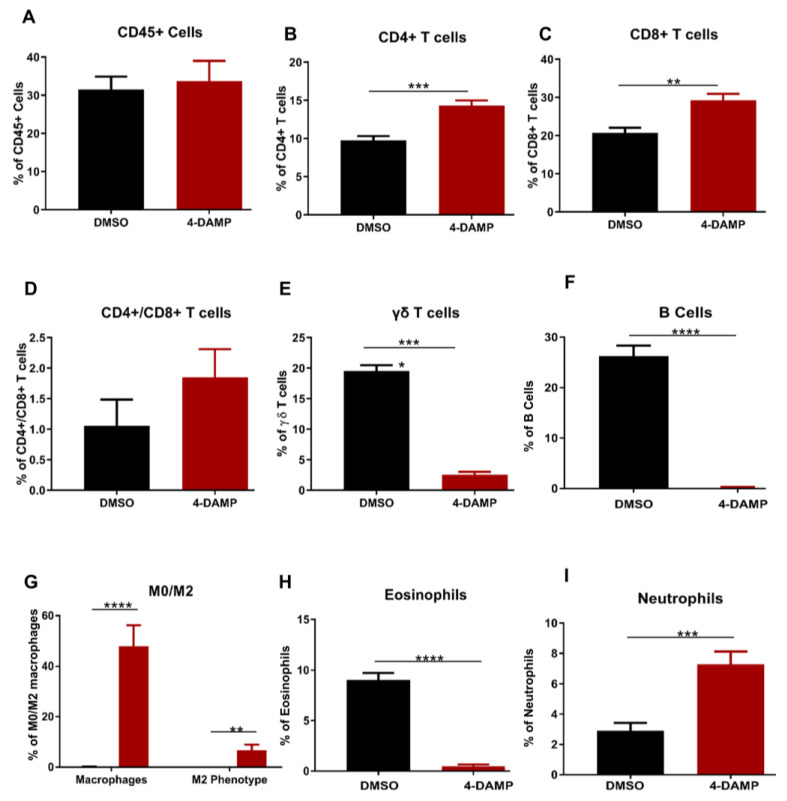
Flow cytometric analysis of CD45+ cells and T lymphocytes. Proportion of the CD45+ cells (**A**), CD4+ T cells (**B**), CD8+ T cells (**C**), CD4+/CD8+ T cells ratio (**D**), γδ T cells (**E**), B cells (**F**), eosinophils (**G**), macrophages (**H**) and neutrophils (**I**) in tumors from DMSO-treated and 4-DAMP-treated groups. Data presented as mean ± SEM, DMSO-treated (n = 4), 4-DAMP-treated (n = 7) mice per group. Student’s *t*-test, ** *p* < 0.01, *** *p* < 0.001, **** *p* < 0.0001.

**Figure 8 ijms-24-00596-f008:**
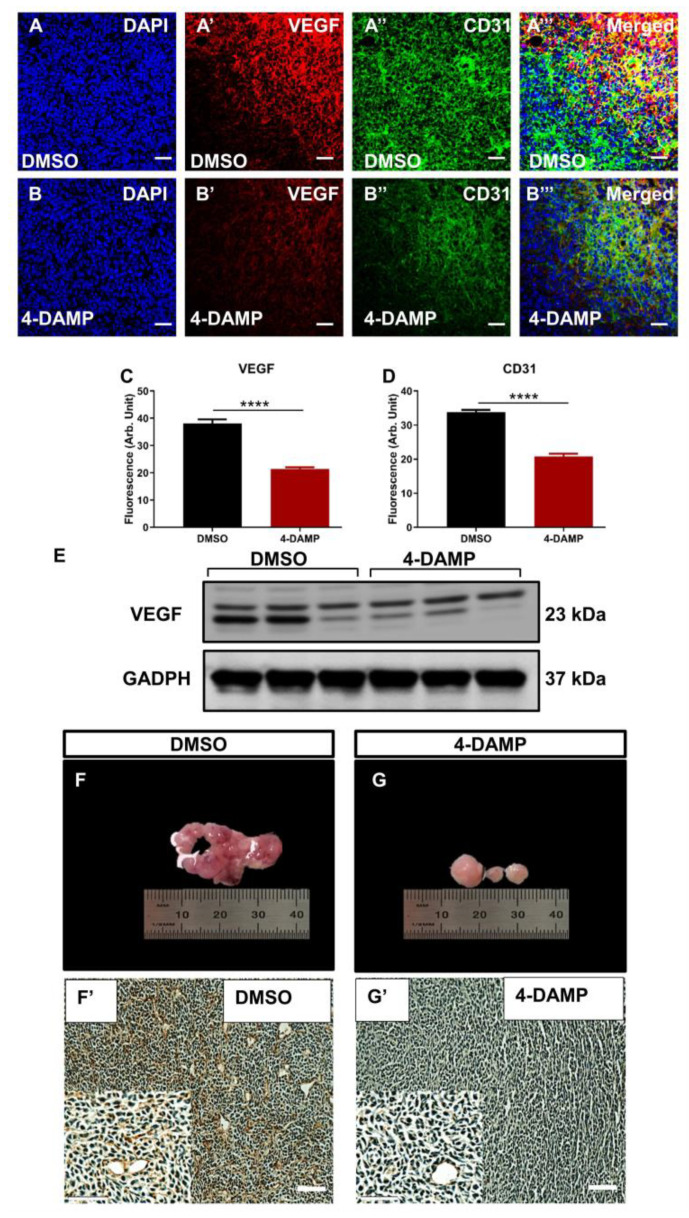
Effect of 4-DAMP treatment on the expression of VEGF and CD31 in vivo. VEGF and CD31 expression in tumor samples from mice bearing CT-26 cell-induced CRC treated with DMSO (**A**–**A’’’**) and 4-DAMP (**B**–**B’’’**). Tumors were labeled with the nuclei marker DAPI (blue; **A**,**B**), VEGF (red; **A’**,**B’**), CD31 (green; **A’’**,**B’’**), and all markers merged (yellow; **A’’’**,**B’’’**). The scale bar represents 50 µm. Bar graphs displaying the mean fluorescence level of VEGF (**C**), CD31 (**D**), and image of VEGF Western blot bands (**E**) in tumor samples from DMSO-treated and 4-DAMP-treated mice. Image showing blood vessels and intensity of CD31 in tumor samples from mice bearing CT-26 cell-induced tumors treated with DMSO (**F**,**F’**) and 4-DAMP (**G**,**G’**). The scale bar represents 100 µm. Data presented as mean ± SEM, n = 8 per group. Student’s *t*-test, **** *p* < 0.0001.

**Figure 9 ijms-24-00596-f009:**
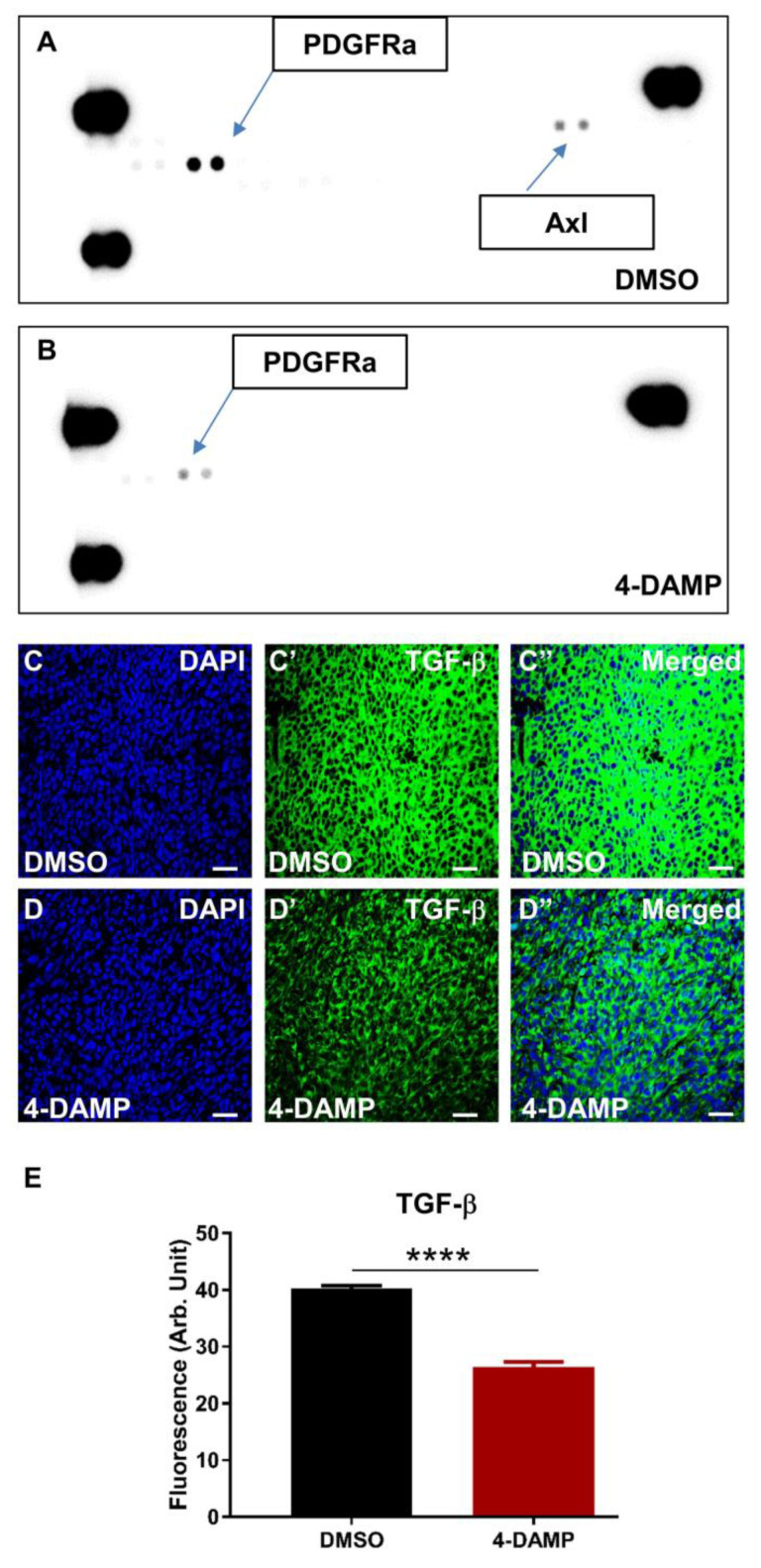
Effect of 4-DAMP treatment on the expression of phosphokinases in vivo. Mouse Phospho-RTK Array measuring phospho-RTK activity in tumors from mice with CT-26-induced CRC treated with DMSO (**A**) and 4-DAMP (**B**). TGF-β in tumor samples from mice bearing CT-26 cell-induced tumors treated with DMSO (**C**–**C’’**) and 4-DAMP (**D**–**D’’**). Tumors were labeled with the nuclei marker DAPI (blue; **C**,**D**), TGF-β (green; **C’**,**D’**), and all markers merged (**C’’**,**D’’**). The scale bar represents 50 µm. Bar graphs displaying the mean fluorescence level of TGF-β (**E**) in tumor samples from DMSO-treated and 4-DAMP-treated mice. Data presented as mean ± SEM, n = 8 per group. Student’s *t*-test, **** *p* < 0.0001.

**Figure 10 ijms-24-00596-f010:**
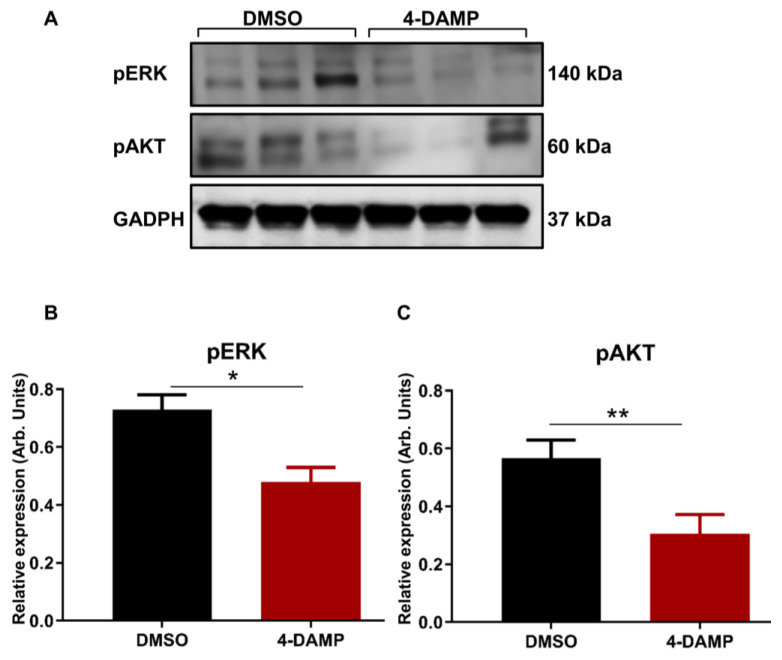
Effect of 4-DAMP treatment on the expression of phosphokinases in vivo. Western blot bands for pERK and pAKT expression in tumor tissues from DMSO-treated and 4-DAMP-treated mice (**A**). Bar graphs displaying the mean intensity of pERK (**B**) and pAKT (**C**) expression in tumor tissues from mice treated with DMSO and 4-DAMP. Data presented as mean ± SEM. Student’s *t*-test, * *p* < 0.05, ** *p* < 0.01.

## Data Availability

The datasets used and/or analyzed during the current study are available from the corresponding author on reasonable request.

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
