# Peer review of "Blocking Muscarinic Receptor 3 Attenuates Tumor Growth and Decreases Immunosuppressive and Cholinergic Markers in an Orthotopic Mouse Model of Colorectal Cancer"

_ijms, 2022, doi:10.3390/ijms24010596_

Round 1
Reviewer 1 Report
This manuscript reports the consequence effects of blocking muscarinic receptor 3 in an orthotopic mouse model of colorectal cancer. Tumor growth, immunosuppressive and cholinergic markers were studied after blocking muscarinic receptor 3 (M3R). The effects of atropine and 1,1-dimethyl-4-22 diphenylacetoxypiperidinium iodide (4-DAMP), the M3R blocker, on cancer growth and spread were evaluated in vitro using murine colon cancer cell line, CT-26. Further investigation in an orthotopic mouse model (in vivo) of colorectal cancer was also investigated. As expected, atropine and 4-DAMP could inhibit CT-26 cell proliferation in a dose dependent manner and induced apoptosis. Atropine could attenuate immunosuppressive markers and M3R via inhibition of EGFR/AKT/ERK signaling pathways. However, 4-DAMP showed no effect on the expression of PD-L1, PD-L2, and ChAT on CT-26 cells but attenuated M3R by suppressing the phosphorylation of AKT and ERK. It was found that blocking M3R with 4-DAMP in an orthotopic mouse model of CRC significantly reduced tumor weight, volume, and size compared to DMSO-treated controls. Overall, this work suggests a crosstalk between the cholinergic and immune systems during cancer development.
This manuscript is well written, and experiments were carefully conducted.
The comments and suggestions are listed below.
1. Abstract; please revise “treatment regimens for CRC patients.” to “treatment regimens for colorectal cancer (CRC) patients.”.
2. Keywords: please replace “CRC” by “Colorectal cancer”. Please use “muscarinic receptor 3” instead of M3R. These abbreviations are the same as other abbreviations, but having different meaning.
Author Response
We are tankful to the Reviewer for their constructive comments. We have addressed all comments and made changes to the manuscript and figures. We hope that these changes improved the quality of this submission.
1. Abstract; please revise “treatment regimens for CRC patients.” to “treatment regimens for colorectal cancer (CRC) patients.”
Response: Thanks for this comment. We made changes to the abstract.
2. Keywords: please replace “CRC” by “Colorectal cancer”. Please use “muscarinic receptor 3” instead of M3R. These abbreviations are the same as other abbreviations, but having different meaning.
Response: Thanks for noticing this. We replaced abbreviations with full definitions.
Reviewer 2 Report
The work presented by Kuol et al, entitled “Blocking muscarinic receptor attenuates tumor growth and decreases immunosuppressive and cholinergic markers in an orthotopic mouse model of colorectal cancer”, describes how the blocking of muscarinic ACh receptors (mAChRs) by using atropine and a 4-DAMP (selective M3R blocker), can reduce tumor growth in colon cancer, acting at the level of immunosuppressive factors (PD-L1 and PD-L2) and angiogenic factors, (such as VEGF). The study is simple but very direct, easy and enjoyable reading, but I would like to ask the authors for several points that I consider key for interpreting the results:
* In the graphs of the effect on cell proliferation (Fig 1 A and C), it would be more appropriated to express the results as % proliferation, to truly assess the effect on proliferation, as in Figure 1B.
* Figure of apoptosis induction (Fig 1 E): the sample treated with Atropine has a very low number of cells, so the results are not representative at all. In Figure 1E we consider that the gates are not well placed, the populations not appearing clearly defined, as it should be in this representation: the double positive population for Annexin and IP is greatly oversized.
* The authors used only one membrane incubated with GADPH, but it seems that there are 4 different membranes, so each one must have its GADPH in order to make the correct quantification. Membranes for in vitro studies should be made in triplicate for each protein. All the WB must be provided to be assessed by the reviewers.
* In in vivo studies, it would be interesting to add weight of the mice throughout the treatment, in order to have data on the potential toxicity of this treatment.
* There is an error in the sentence: "To compare the effects of atropine and a selective M3R blocker, 4-DAMP cells...". "4-DAMPcells", should be replaced by "CT-26 cells"
* Check the scale bars to normalize them: in some images they are different within the same sample, eg. Fig 8D
Author Response
We are tankful to the Reviewer for their constructive comments. We have addressed all comments and made changes to the manuscript and figures. We hope that these changes improved the quality of this submission.
* In the graphs of the effect on cell proliferation (Fig 1 A and C), it would be more appropriated to express the results as % proliferation, to truly assess the effect on proliferation, as in Figure 1B.
Response: Thank you for your comment. We have made changes to Fig 1A and 1C, expressing the results as % proliferation.
* Figure of apoptosis induction (Fig 1 E): the sample treated with Atropine has a very low number of cells, so the results are not representative at all. In Figure 1E we consider that the gates are not well placed, the populations not appearing clearly defined, as it should be in this representation: the double positive population for Annexin and IP is greatly oversized.
Response: Thank you for your comment. We have made changes to Fig 1E, we have provided representative images. All gating and compensation were made consistent for all groups.
* The authors used only one membrane incubated with GADPH, but it seems that there are 4 different membranes, so each one must have its GADPH in order to make the correct quantification. Membranes for in vitro studies should be made in triplicate for each protein. All the WB must be provided to be assessed by the reviewers.
Response: We provided all WB images as supplementary materials. Unfortunately, all in vitro Western Blots were run as a single well, but experiments were repeated 3 times. We will keep in mind to always run triplicate well. Regarding GAPDH, it was not possible to run two proteins with close molecular weight on one membrane and we did not have a stripping buffer. However, all membranes were run on the same day.
* In in vivo studies, it would be interesting to add weight of the mice throughout the treatment, in order to have data on the potential toxicity of this treatment.
Response: Thank you for the suggestion. We have added data on the weight of the mice throughout the treatment (please see revised Figure 4A).
* There is an error in the sentence: "To compare the effects of atropine and a selective M3R blocker, 4-DAMP cells...". "4-DAMPcells", should be replaced by "CT-26 cells"
Response: Thank you for noticing this error. We have replaced “4-DAMP cells” with “CT-26 cells”.
* Check the scale bars to normalize them: in some images they are different within the same sample, eg. Fig 8D
Response: Thank you for noticing this. We have checked all images and provided the correct scale bars.